# Evolution of Physical Fitness in Formative Female Basketball Players: A Case Study

**DOI:** 10.3390/sports8070097

**Published:** 2020-07-08

**Authors:** David Mancha-Triguero, Nicolás Martín-Encinas, Sergio J. Ibáñez

**Affiliations:** Grupo de Optimización del Entrenamiento y Rendimiento Deportivo (GOERD), Faculty of Sport Sciences, University of Extremadura, 10003 Cáceres, Spain; nikimartien28@gmail.com (N.M.-E.); sibanez@unex.es (S.J.I.)

**Keywords:** physical fitness, game position, test, integral assessment, microtechnology

## Abstract

Over the last few years, team sports increased the amount of physical demand and its importance. Therefore, work related to physical fitness and its assessment is essential to achieving success. However, there are few studies on this subject at the formative stage. The purpose of this study was then to analyze the physical fitness of an under-18 (U18) women’s team divided by game positions. In addition, physical fitness at different times of the season was characterized to identify differences and determine its evolution. To assess physical fitness, tests of aerobic and anaerobic capacities, lower body strength, centripetal force, agility and speed were carried out as designed in the SBAFIT battery. Each player was equipped with an inertial microtechnology device for the collection of data. This research is classified as empirical, with quasi-experimental methodology. The results showed significant differences in variables of the test of aerobic and anaerobic capacities, speed, agility (generic and specific), and centripetal force (right) based on game position and the moment of the season. The results also showed the importance of the specific physical aspect in relation to an optimal improvement in physical fitness, since training sessions and competition do not allow all players to improve equally or efficiently.

## 1. Introduction

Basketball is a complex and dynamic team sport that combines explosive movement structures with different technical skills, such as bouncing, passing, or shooting to the basket [1]. Success in basketball depends on the sum of several factors [2], including morphological attributes [3], physical and technical skills [4], and tactical actions [5]. Therefore, achieving success requires a meticulous and individualized process in which competition plays a fundamental role. The analysis of competition provides great knowledge for planning the training since, during competition, athletes reach their maximum level and performance; thus, these values can be taken into account as a reference to adapt and plan the training [6].

Regarding the physiological aspects of the sport, basketball can be defined as a hybrid sport [7] as, during practice, athletes alternate moments in which energy is obtained through aerobic metabolism and others in which energy is obtained through the anaerobic pathway. This continuous variation of the energy source depends on the action or moment in the match and, consequently, it affects the physical performance of the athlete. Physical performance is of great importance since it is directly related to performance in competition [8]. Furthermore, along these lines, Reference [9] confirmed that physical fitness (PF) is related to performance indicators in competition and, therefore, a better level of PF helps to achieve both individual and collective performance. The evaluation of PF can be done through different physical aptitude tests. These tests vary widely and sometimes provide antagonistic results, since the tests used may not take into account the level, the sport, or the category [10].

The analysis of existing tests in the literature to evaluate the PF of athletes can be grouped according to specificity and origin into two large groups. On one hand, taking into account the specificity of the test, there are general tests and specific tests for PF. The main characteristic of general tests is that they compare samples belonging to different sports, and they do not require that the person in charge of carrying out the test has specific knowledge of the sport itself [11]. As for specific tests, they are characterized by simulating the requirements that the athlete faces during competition, as well as taking into account formal aspects of the game. These tests, rarely used in the literature, are ideal for evaluating athletes, taking into account the different contextual variables that can affect performance (category, level, gender, etc.) [12]. The specificity of the test with respect to the sport helps the athlete to obtain more reliable and valid results than those provided by general tests [11]. On the other hand, regarding origin, the tests can be differentiated as laboratory tests and field tests. While laboratory tests allow for a more objective measurement, field tests offer more ecological results. In line with this, the principle of specificity favors the athlete obtaining better results with the most familiar activity (field test) [12].

The assessment of PF can be carried out for different purposes, such as monitoring, control, or validation, allowing an evaluation and comparison of the PF of athletes at different times of the season. This information helps to determine the methods and means to be adapted, thereby optimizing training and consequently improving the athlete’s performance [13] or recovering the player’s level after a period of inactivity because of injury [14], while it may also be used as a talent detection method [15]. The importance of subjecting the athlete to continuous assessments is due to the fact that PF is a changing aspect in athletes [16], being a limiting factor of performance. Moreover, due to training, a set of adaptations is produced which can affect planning [6]. In line with this, some research studies affirmed that multiple assessments must be performed during the season (depending on the moment or the objectives of the season) and that the results of these assessments are meaningful when planning both training sessions and competitions [6].

Historically, the most widely used indicator of performance in invasion sports is individual statistics [17]. The results of different studies that analyzed individual technical–tactical performance indicators such as team performance factors were influenced by different contextual situations such as the final score [18], sex [19], level of competition, age [20], and physical characteristics of the team [21]. For all these reasons, some authors preferred using physical attributes as a performance factor in basketball for the individual improvement of each player, without taking into account the performance of the team [22].

Scientific literature on women’s basketball is still limited [23]. The most common line of research in female basketball players is related to injury. Additionally, women’s sport is often studied through hormonal, biological, and anatomical factors that define the characteristics of players, but not through an analysis of how to work with them. Along these lines, Reference [24] presented female athletes as a unique challenge for sports medicine, since they run a greater risk than male athletes of suffering injuries that are related to their morphological and physiological differences. To solve this, it is important to investigate training and competition in this population whose physical condition is often characterized from male data [25]. Therefore, to optimize performance in women’s basketball, it is necessary to respect the principles of sports training, such as individuality and specificity [26].

Based on the above, there are few documents related to the evaluation of PF in an integral way for women in basketball, while there were different studies in the male category at different ages and levels [27,28]. Therefore, it is very difficult to extrapolate the results to female basketball in general and to formative stages in particular. The research carried out so far does not provide a clear understanding of physical work since there were contradictory results. For all these reasons, the objectives of this study were (i) to characterize the PF of a women’s team in the under-18 (U18) category, (ii) to analyze the differences in PF by game position, and (iii) to identify the differences in the evolution of PF by game positions.

## 2. Methods

### 2.1. Design

This research can be defined as empirical with an associative strategy; it seeks to examine the differences in variables analyzed from the same group, in order to compare the physical fitness of players at different times of the season. It can then be considered an evolutionary prospective design [29].

### 2.2. Participants

The selected sample was an under-18 category female team (*n* = 10; 17 ± 0.82 years, a weight of 57.3 ± 5.7 kg, a height of 168.71 ± 8.29 cm, a wingspan of 166.81 ± 5.81 cm, and basketball experience of 8.56 ± 1.02 years). The analyzed team played the last six seasons in regional competitions in different categories and was the winner in several of them. Furthermore, the team was the regional winner in the season analyzed; therefore, they competed in the National Championship representing their autonomous region. The weekly training time was divided into three days (2 h per day) and a match day (always on weekends). Out of the 10 players who form the team, three occupy the guard position, four are forwards, and three occupy the center position. Three PF assessments were carried out at different times of the season (September–April) (statistical analysis units = 30). The coaching staff and the players were previously informed of the details of the study and they gave informed consent to participate. In the case of minor players, the consent form was signed by their legal guardians. The study was developed according to the ethical provisions of the Declaration of Helsinki (2013) approved by the Bioethics Committee of the University (number 233/2019).

### 2.3. Variables

In this research, the moment of the season and the game position were considered as independent variables. To assess the PF of the players, we analyzed the following variables that were divided into five groups based on the type of demand [30]: (i) technical–tactical variables, (ii) objective internal load variables, (iii) objective external load kinematics variables related to distance or time, (iv) objective external load kinematics variables related to accelerometry, and (v) objective external load neuromuscular variables. The selected variables were defined and used in other studies whose research topic was in line with the objectives of this research [31,32,33].

(i)*Technical–tactical variables* analyze the technical gesture of the shot to basket using an observational methodology during the aerobic capacity and anaerobic capacity test.
*Shots* makes reference to the number of shots that the player makes during the test.*Scores* refers to the number of shots that are finally scored.*Efficacy (%)* is the value (expressed in %) given from the division between scores and shots. There is a document that offers guidance values on these variables according to the player’s age and gender [34].(ii)*Objective internal load variables* are assessed through the heart rate (HR) and make reference to the player’s demands during a task or training sessions. Within these variables, the following parameters are analyzed:
*Heart rate maximum (HR Max)* is the maximum value of beats per minute reached by the player during the test.*Heart rate medium (HR Avg)* is the average value of beats per minute during the test.*Heart rate recovery (HR Rec)* is the value of beats per minute two minutes after the end of the test when the player is advised to do passive recovery [34].(iii)*Objective external load kinematics variables related to distance or time* analyze the external load supported by the athletes by means of distance or time in relation to the execution time and their movements.
*Part of circuit* refers to the number of circuit fragments completed by the athlete during the test. Unlike the aerobic test in which each circuit is formed by 12 fractions, the anaerobic test is formed by four fractions [34].*Time* is the period that the athlete uses to move from one point to another (measured in seconds).(iv)*Objective external load kinematics variables related to accelerometry* register the external load supported by the athletes by means of accelerometry in relation to the execution time and their movements.
*Accelerations* refers to the positive increase in speed (total and per minute).*Decelerations* is the negative increase in speed (total and per minute).(v)*Objective external load neuromuscular variables* analyze the external load that the athlete receives in proportion to the gravitational force. Within this category, two variables are analyzed:
*Impacts* are assessed through the force that the musculoskeletal structures bear in proportion to the gravitational force (G-force).*Player load* (PL) is a vectorial magnitude derived from triaxial accelerometry data that quantifies the movement at a high resolution. In order to obtain a cumulative measure of the rate of change in acceleration, both accelerations and decelerations are used. A cumulative measure (PL) and a measure of intensity (PL·min^−1^) are analyzed, which can provide the stress rate to which players subject their body for a certain period of time [35].

The variables accelerations, decelerations, and player load were normalized per minute for the reliability and equality of data.

### 2.4. Materials and Instruments

For the recording of technical–tactical variables, a video camera (JVC model GY-HM70U) and a record sheet were used to register the number of shots and the score or error sequence. To record objective internal load variables, each athlete was equipped with a GARMIN^®^ (Kansas, USA) heart rate band. Finally, for the registration of objective external load kinematics variables related to distance and those related to accelerometry, as well as objective external load neuromuscular variables, each player was equipped with an inertial device model WIMU^®^ (RealTrack Systems, Almería, Spain), which was fixed using an anatomically adapted harness for each player. Furthermore, to analyze the data regarding time, ChronoJump^®^ photoelectric cells (Bosco System, Barcelona, Spain) were used. Once recorded, we analyzed the data by means of the SPRO^®^ software (RealTrack Systems, Almería, Spain). The instrument used to assess the athletes’ overall PF was the SBAFIT test battery [16]. This battery assesses different capacities including (i) aerobic capacity, (ii) lactic anaerobic capacity, (iii) maximum strength of the lower body (Abalakov test), (iv) reactive strength of the lower body (multi-jump test), (v) travel speed (Repeat Sprint Ability test 5 × 14 m), (vi) agility (T test generic and specific), and (vii) centripetal force (arc test right and left). The agility test was carried out with two variants: a generic one (athlete’s displacement) and a specific one (athlete’s displacement while bouncing the ball).

### 2.5. Statistical Analysis

Firstly, a descriptive analysis of all the variables in each of the tests of the test battery (*mean* and *SD*) was performed, dividing the sample by game position (guard, forward, and center). Secondly, we carried out an exploratory analysis using the assumption of criteria tests [36]; as we found a normal distribution of data, parametric tests were carried out to test the hypotheses. Subsequently, one-way *ANOVA* with *Bonferroni post hoc* was used to identify the differences between the different game positions in every evolution assessment and in the global assessment [36]. Upon dividing the sample by game position in each moment of the season, a descriptive analysis (*mean* and *SD*) was performed to determine the evolution at different times of the season. Afterward, the *general linear model* of repeated measures was used to identify significant differences between the different moments of the season. *Mauchly’s W* sphericity test was taken into account, bearing in mind that the sample size was small. The sphericity hypothesis could not be rejected; thus, the assumed sphericity value was used to determine if the differences were significant [37,38]. Finally, for the presentation of the descriptive results, the results were normalized through the *Z-score*. The *Z-score’s* purpose is to standardize a value so that it represents the standard deviation that the value is above the mean [39]. The results were presented in profiles based on the game position of the analyzed players.

### 2.6. Procedure

Firstly, the club and the coaches were contacted to be informed about the project. After the proposal was accepted, parents were given an informed consent form with relevant information of the research. Secondly, once the team’s competitive calendar was analyzed, the moments when there was no competition were selected in order to do the fitness tests. This was done so as to make sure that the athletes were in the best PF possible and so that the fatigue produced did not affect any subsequent competitions. The three different moments of the season selected were due to competition absences (the third week of December coinciding with the end of the first competitive macrocycle, the fourth week of February coinciding with the second competitive macrocycle, and the fourth week of April coinciding with the preparation for the National Championship). After collecting the data, the coach was given a dossier with the information obtained from the tests so as to have more information about the PF of the players. We followed the protocol described in the SBAFIT test battery when doing the tests [16]. The tests were carried out on two different days with a difference of at least 72 h of recovery so that the players were fully recovered and the results were more reliable. In the end, all participants in the study carried out two training sessions with the material to be used in the measurement when doing the tests. The purpose of having a contact was to avoid ignorance or discomfort without affecting performance.

## 3. Results

Figure 1 and Table 1 show the descriptive results of the tests analyzed, differentiating the players by game position at different times of the season.

As observed in Figure 1, the evolution of the PF of the players according to the moment of the season and game position varied depending on different requirements such as competition and training. Furthermore, there were significant differences in the vast majority of tests performed. In the aerobic capacity test, it was observed that the results showed significant differences in the first assessment with guards compared to centers and forwards. These differences disappeared in the following assessment, and the results between game positions were homogenized. In the anaerobic capacity test, the results showed that, in the first assessment, there were significant differences between game positions. The guards showed significant differences with regard to forwards and centers. In the second assessment, these differences disappeared; however, in the third assessment, coinciding with the end of the season, they appeared again as in the first assessment. In the Abalakov test and multi-jump test, the results showed significant differences in the first assessment between forwards and centers. These differences in the second assessment changed, and significant differences between guards and forwards appeared, while, in the multi-jump test, there were no differences between game positions. Finally, the results were equal in the last assessment, and no significant differences were found according to game position. In the RSA test, the results showed significant differences in the first assessment between guards and centers. These differences in the second and third assessments disappeared, and the results were homogenized without taking into account the game position. In the T test (generic and specific), the results did not show significant differences between game positions. In the second assessment in the generic test, significant differences appeared between forwards and centers. Finally, in the third assessment of the generic test, significant differences appeared between guards and forwards, while, in the specific test, no significant differences were found throughout the season. Finally, in the Arc test, the results were similar throughout the season. There were no significant differences between game positions in any of the assessments. In both the right-hand test and the left-hand test, the differences between players were small.

Table 1 complements Figure 1, since all the variables of each test were analyzed and the different assessments, and the significant differences between game positions are shown. Table 1 shows the results obtained based on the different game positions in each moment of the season. The results showed significant differences in the tests of aerobic capacity, travel speed (RSA), generic agility, specific agility, and left centripetal force depending on the game position taking into account the three assessments carried out during the season. The guards and centers were those that obtained the greatest significant differences, with their evolution being higher than that of the forwards. As observed in Table 2, the results obtained from the analysis of the three assessments together showed significant differences in the majority of the analyzed tests. In the aerobic capacity test, the results between guards, forwards, and centers were similar, although there were significant differences in the variables *Scores* (*p* = *0*.038), *Efficacy* (*p* = *0*.031), and *Impacts* (*p* = *0*.041). In the anaerobic lactic capacity test, the forwards were those with the best level, while the guards were those with the worst results in the test. In addition, there were significant differences between positions in the variables *Parts of circuit* (*p* = *0*.023), *Shots* (*p* = *0*.019), and *Heart rate average* (*p* = *0*.036). In the lower body strength tests (Abalakov and multi-jump), the guards were those who obtained the best results in both, accompanied by centers in the multi-jump test. Regarding the displacement speed test (RSA test), the forwards were those that obtained the best results with significant differences in the *Impacts* variable (*p* = *0*.034). In the agility tests, the centers and forwards obtained the best results. Furthermore, in the generic test, there were significant differences in the *Heart rate maximum* (*p* = *0*.042), *Impacts* (*p* = *0*.001), *Player load* (*p* = *0*.031), and *PlayerLoad*/*minute* (*p* = *0*.001), while, in the specific test, there were significant differences in the *Heart rate maximum* (*p* = *0*.041) and *Average heart rate (p* = *0*.033). Finally, in the centripetal strength tests, all players obtained similar results in the execution time, although, in the test to the right, there were significant differences in the *Impacts* variable (*p* = *0*.012).

## 4. Discussion

The objectives of the present study were to analyze the differences in PF by game position and to identify the differences in the evolution of PF in different moments of the season of a U18 female basketball team. In addition, an analysis was carried out between game positions with general differences in the global assessment. Once the analysis was done, some differences were found depending on the game position and for different assessments of the season moment in which the tests were carried out.

The analysis of the internal and external load using an integral battery of basketball-specific field tests is not common practice in women’s sports and, to a lesser extent, in formative sports. Currently, the assessment of PF is carried out through laboratory or generic tests regardless of the sport practiced [12]. Due to the lack of knowledge in this type of sample, the results are sometimes adapted from results that come from high-level teams [40] or national teams [41]. For this reason, it is believed that the analysis of PF in training categories is relevant since the principles of specificity and individualization of the athlete are ignored [26].

### 4.1. Differences by Game Positions

The data obtained in this study show that the players with center positions are those that obtained the highest scores and efficiency when shooting in the aerobic test. In line with this, Reference [42] analyzed the statistics in European competitions for game positions and observed that the centers were those who scored the most two-points shots. Similar results were observed in the rest of the analyzed variables, regardless of the game position. Along these lines, Reference [43] considered that the aerobic results of the athletes should be homogeneous regardless of the position of the players. In contrast to what was stated previously, Reference [44] confirmed significant differences between game positions in the same category by carrying out generic tests. The results obtained in this study coincide with those existing in the literature in that the centers are those who make the best two-point shots and there are no major differences in the aerobic capacity between different game positions. These statements may be due to the fact that, because of their game being closer to the basket, the centers are used to throwing from that area regardless of the game category.

With regard to the Anaerobic Capacity test, the guards were those who completed the fewest *Parts of circuit*. According to Reference [43], there are significant differences in this ability for players in the same category. In addition, Reference [45] added that these differences may be a predictor of the level of game. This could be due to the fact that forward and center players in this team have higher levels than the guards. The test that Reference [45] carried out was one in which the athlete travels without taking into account elements or technical–tactical actions of the sport.

In the lower body strength tests, no significant difference was observed. All players showed similar lower body strength and lower body fatigue tolerance. So far, the studies that performed the jump test in players in the U18 category mostly involved male teams. Despite this, the literature shows very disparate results [44,46,47]. This variety in the obtained results is characterized by having samples of different genders or competitive levels to the team analyzed in this study. In this research, the results obtained compared to those found in the literature are similar, considering that they were compared with male players. For this reason, it can be stated that, although there are differences between genders and game positions, the differences are usually similar regardless of the gender of the players.

In the displacement speed test, significant differences were observed in the impacts of centers, which may be due to their larger size. Regarding the time variable, no significant differences were observed between game positions. This may be due to the fact that there is not much anthropometric difference between centers and other players in different game positions. Although there are no differences in the time used, the centers support more load and fatigue before the same stimulus (14-meter sprint). Regarding the distance to travel, each research study in the literature chose a different distance to that chosen in this research.

Agility tests showed differences in time for the generic and specific tests. In addition, in the specific agility test, the results showed that forwards obtained lower *HR Max*. In terms of generic agility, forwards showed significant differences in the *Impacts* variable, related to PL and PL/min. In line with this, Reference [46] carried out a test with and without the ball, finding differences in time between both tests due to the involvement of the ball. With regard to agility and direction changes, Reference [48] found a correlation between the T test and the changes in direction that players make during the game. In this study, the execution time of the test was higher than the values that exist in the literature. These differences may be due to the level of competition and to the comparison between male and female teams. Regarding the differences between generic and specific tests, the PF is the only aspect evaluated in the generic test. However, in the specific test, the technique of the evaluated player affects the physical aspects due to them intrinsically carrying the ball in the displacements. Furthermore, the existing research in the literature used male players as participants.

As for the changes of direction, the centripetal force test forces the athlete to undergo continuous changes of direction so that they can move at the highest possible speed with a curved trajectory. The results showed significant differences in the centers, who performed a higher number of impacts. Regarding this test, there are no similar tests in the literature that assess the athlete’s centripetal force as it is a new assessment. The differences of centers, e.g., in the displacement speed test, are due to them supporting greater fatigue before the same stimulus than the guards or forwards.

### 4.2. Evolution during the Season

In the aerobic capacity test, the results showed significant differences between the players with the same game position at different times of the season. The results showed that the guards and the centers obtained significant differences in the variables HR recovery, accelerations/minute, and decelerations/minute. In addition, players with a forward position obtained significant differences in the player load and player load/minute variables. Along these lines, Reference [49] analyzed the evolution of the aerobic capacity of a U18 team and discovered an improvement over the months. These improvements were due to the training process that improved physical qualities, although the improvements were not optimal. The results obtained in this research showed a significant improvement throughout the season, possibly due to the accumulation of training load supported by the players at the end of the season.

In the anaerobic capacity test, no significant differences were found during the season. Related to these findings, Reference [50] demonstrated that training and competition processes improve this capacity during the season. Contrary to these findings, the results of this research confirmed that there was no improvement in this capacity during the season. These results may be due to the training process that did not include specific tasks to improve this ability. The differences between the findings of this research and the results existing in the literature may be due to the fact that they were different sports although they shared formal logic (invasion sport and intermittent), as well as the fact that the participants were male players.

There were also no significant differences identified during the season in the lower body strength tests. Related to this capacity, Reference [49] stated that the lower body strength, frequently used for jumps and different movements in sports like basketball, undergoes an evolution during the season as a result of training and competition. In this case, the study sample did not show improvements in this capacity, and the cause may have been the same as for anaerobic capacity. Related to the lower body strength, the results in the displacement speed test (RSA test) showed significant differences in the guards and centers in the time variable and player load. Along these lines, Reference [51] demonstrated that this variable showed differences during the season. These differences allowed the athletes to improve their values at the beginning and, as a result of training and competition, they reached a plateau where this quality did not improve and, finally, the results worsened as a result of fatigue and the high end-of-season load. This may be due to the fact that, during the training process, no specific tasks were designed to work this capacity or no extra work was done to improve this capacity.

As for the agility tests, in the generic version, there were significant differences between positions in the variables time and player load. This is because, if an athlete maintains demand for a longer time, the total value of this demand (PL) will also be higher. By contrast, in the specific version, there were also significant differences in the player load variable between players. Related to this, Reference [49] analyzed the evolution of agility. The results showed the same pattern as in the RSA test.

Finally, in the centripetal strength tests, significant differences were only found in the variable time for the left test in the guards. There is a gap in literature related to this capacity, which became fashionable in recent years [16]. The centripetal force is a quality of the athlete related to the ability to generate changes of rhythm and direction, commonly used in team sports.

## 5. Conclusions

The results obtained in the present study of the analysis and differentiation of PF in female U18 basketball players showed that there are differences in the level of PF of the players with different game positions. These results showed values in the PF tests regardless of the game position and the anthropometric factors linked to it. The differences in the evolution of PF during the season were analyzed according to the game position. The results showed little evolution of the PF during the season. These differences were found in all tests performed except for three tests (agility specific, centripetal force right, and centripetal force left). As for the differences between game positions, most of them occurred between guards and forwards. Finally, the test that provided the greatest difference between positions was the general agility test. These differences may be due to the fact that, in this test, only the PF of the athlete is evaluated, without taking into account other aspects as in the specific agility test (technical aspects of the sport itself). For all these reasons, it can be stated that the training designed by the coach is not optimal since it does not take into account the demands and requirements of U18 female basketball players. For further research, it would be interesting to replicate the approach with a larger number of samples, categories, genders, and geographic areas.

### 5.1. Practical Applications

The results of the research require new strategies to improve and maintain the PF of the analyzed players over time in order to optimize performance in competition. For this, some of the applications to be carried out may be (i) to modify the training structure since not all the players improved their PF, (ii) to design specific tasks of PF [52] for some qualities that are very important in the match but that during training do not appear as much as in competition, (iii) to introduce specific training sessions to work mainly on strength (strength task in the gym and strength task in the field adapted to basketball), where it is advisable to design training sessions at least once a week to work on the strength or power depending on the moment of the season and the objectives to be achieved, (iv) to perform a physical periodization to train during the season and, based on that, to design tasks with more or less rest and different intensities [53], and (v) to work on physical aspects, individually taking into account the requirements of the competition or training sessions, the specific game position, and the results obtained in physical fitness assessments. For example, guards and forwards must perform more tasks related to explosive strength, agility, and speed, while centers should perform tasks focused primarily on gaining strength.

### 5.2. Limitations

This study analyzed a single regional level team that competes in the Spanish Championship, but that does not qualify for the final phase. Therefore, the obtained results are not generalized, since the physical and anthropometric conditions of the players are a factor to be taken into account in this type of research. In addition, by collecting data from only one team, the results showed the casuistry of that team, which may not be the same in another team. This paper sought to show the importance of monitoring and assessing the PF of athletes to individualize training based on the needs and specificities of each player.

## Figures and Tables

**Figure 1 sports-08-00097-f001:**
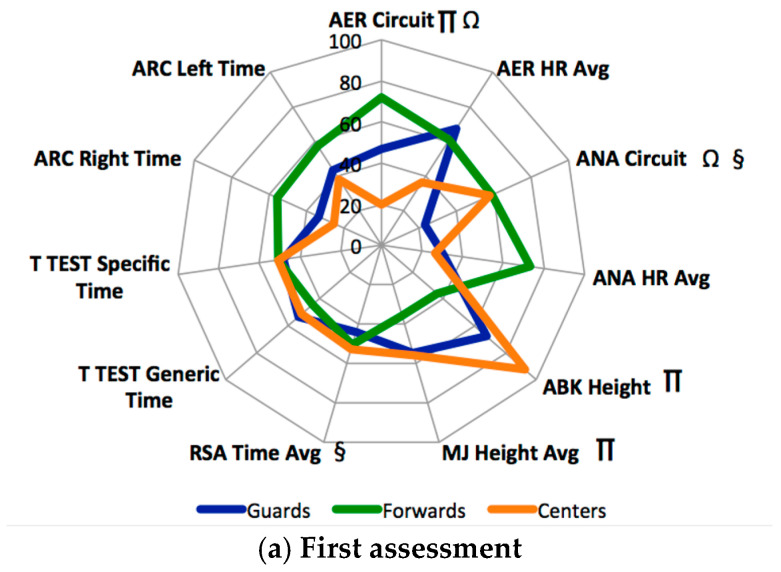
Standardized descriptive results based on game position. ***AER Circuit***: parts of circuit carried out by the players in aerobic capacity; ***AER******HR Avg***: heart rate average in aerobic capacity; ***ANA***
***Circuit**:* parts of circuit carried out by the players in anaerobic capacity; ***ANA******HR Avg***: heart rate average in anaerobic capacity; ***ABK******Height***: height achieved by players in Abalakov test; ***MJ***
***Height***: height achieved by players in multi-jump test; ***RSA***
***Time Avg**:* time average in RSA test; ***ARC******Right**:* right centripetal strength test; ***ARC Left***: left centripetal strength test; ***Ω***: significant post hoc differences between guards and forwards; ***§***: significant post hoc differences between guards and centers; ***∏***: significant post hoc differences between forwards and centers.

**Table 1 sports-08-00097-t001:** Descriptive results of the analyzed and inferential variables depending on the moment of the season.

		Assessment 1			Assessment 2			Assessment 3					
		G	F	C	G	F	C	G	F	C	*Mauchly’s W*		
		*Avg* ± *SD*	*Avg* ± *SD*	*Avg* ± *SD*	*Avg* ± *SD*	*Avg* ± *SD*	*Avg* ± *SD*	*Avg* ± *SD*	*Avg* ± *SD*	*Avg* ± *SD*	G	F	C
Aerobic Test SIG/AER	Circuit	128.33 ± 8.74	138 ± 7.59	122.5 ± 4.71	135.33 ± 8.08	126 ± 19.8	145.5 ± 9.19	131.67 ± 2.52	132.5 ± 4.95	129.5 ± 0.71	0.079	0.254	0.717
	Shots	11 ± 1	12.5 ± 0.71	10.5 ± 0.71	11.33 ± 0.58	11 ± 1.41	12.5 ± 0.71	11 ± 0	11 ± 0	11 ± 0	0.840	0.125	0.071
	Scores	5.33 ± 3.06	3 ± 1.41	6 ± 1.41	3 ± 1	3.5 ± 2.12	8 ± 1	4 ± 4.36	2.5 ± 0.71	5.5 ± 2.12	0.700	0.563	0.250
	Efficacy (%)	47.07 ± 24.22	25 ± 11.78	56.82 ± 9.64	26.51 ± 9.18	30.84 ± 15.32	64.11 ± 3.63	36.36 ± 39.63	22.73 ± 6.43	50 ± 19.29	0.715	0.639	0.571
	HR Avg	190 ± 5.29	193 ± 14.14	173 ± 5.66	160 ± 36.37	171.5 ± 14.85	181.5 ± 2.12	183.33 ± 3.51	185 ± 1.41	181 ± 14.14	0.141	0.476	0.686
	HR Max	199.33 ± 7.77	199.5 ± 16.26	194 ± 8.49	170.67 ± 37.82	189.5 ± 0.71	198.5 ± 6.36	194.33 ± 4.51	190.5 ± 14.85	194.5 ± 17.68	0.385	0.552	0.748
	HR Rec	149.33 ± 4.51	139.5 ± 19.09	147.5 ± 4.95	121.33 ± 7.23	121 ± 12.73	132 ± 5.66	123.67 ± 5.03	126.5 ± 7.78	135.5 ± 9.19	0.007 *	0.191	0.038 *
	Impacts	464.67 ± 130.99	491 ± 154.97	624 ± 133.57	410.67 ± 132.16	429.5 ± 171.23	663 ± 60.81	464.67 ± 175.51	436 ± 102.64	683 ± 152.14	0.996	0.422	0.556
	PL	27.27 ± 2.85	28.04 ± 1.53	28.25 ± 1.73	25.14 ± 0.8	25.04 ± 1.93	27.47 ± 1.29	27.14 ± 4.84	25.05 ± 1.8	27.81 ± 1.62	0.797	0.003*	0.915
	PL/min	2.28 ± 0.24	2.34 ± 0.13	2.35 ± 0.14	2.1 ± 0.06	2.09 ± 0.16	2.29 ± 0.11	2.26 ± 0.4	2.09 ± 0.15	2.32 ± 0.13	0.795	0.003 *	0.914
	Acc/min	34.66 ± 0.93	33.83 ± 0.55	34.79 ± 1	28.01 ± 3.58	29.37 ± 4.79	29.44 ± 3.5	37 ± 1.15	36.75 ± 2	37.46 ± 1.83	0.028 *	0.152	0.046 *
	Dec/min	34.66 ± 0.92	33.78 ± 0.63	34.79 ± 1	27.99 ± 3.57	29.41 ± 4.73	29.4 ± 3.56	37 ± 1.22	36.75 ± 2	37.45 ± 1.71	0.031 *	0.151	0.049 *
Anaerobic Test SIG/ANA	Circuit	110.67 ± 1.53	117.5 ± 2.12	121.5 ± 12.02	115.33 ± 3.79	120 ± 5.66	118 ± 5.66	113 ± 3.46	118 ± 2	117.5 ± 0.71	0.411	0.718	0.810
	Shots	29.33 ± 0.58	29.5 ± 0.71	32.5 ± 2.12	29.67 ± 0.58	30 ± 1.41	31 ± 1.41	29.33 ± 1.15	30 ± 0	30 ± 0	0.975	0.250	0.269
	Scores	24 ± 6.24	27 ± 2.83	28 ± 5.66	25 ± 4.58	24 ± 2.83	28 ± 1	26 ± 3.61	22.5 ± 2.12	25.5 ± 6.36	0.679	0.321	0.745
	Efficacy	81.65 ± 20.25	91.44 ± 7.4	85.77 ± 11.81	84.33 ± 15.69	79.87 ± 5.66	90.42 ± 4.12	88.41 ± 9.15	75 ± 7.07	85 ± 21.21	0.670	0.344	0.905
	HR Avg	169.67 ± 5.13	183 ± 7.07	171.5 ± 0.71	177 ± 13.11	182 ± 12.73	163.5 ± 4.95	169.33 ± 4.62	173.5 ± 2.12	167 ± 7.07	0.589	0.485	0.529
	HR Max	191.67 ± 3.79	198.5 ± 13.44	193.5 ± 3.54	194 ± 9.54	195.5 ± 12.02	189.5 ± 7.78	190.67 ± 1.15	187 ± 0	183.5 ± 4.95	0.747	0.434	0.084
	HR Rec	133.33 ± 4.62	138 ± 1.41	129.5 ± 4.95	131.33 ± 29.87	135.5 ± 10.61	136 ± 19.8	126 ± 3.46	137.5 ± 2.12	130.5 ± 7.78	0.265	0.164	0.862
	Impacts	240.67 ± 111.63	199 ± 5.66	257.5 ± 33.23	300.33 ± 28.94	224.5 ± 26.16	301.5 ± 26.16	568.67 ± 45.83	334 ± 74.95	352 ± 49.5	0.252	0.118	0.262
	PL	12.39 ± 1.54	12.17 ± 0.21	12.81 ± 0.35	12.72 ± 0.65	11.79 ± 0.14	12.22 ± 0.62	14.21 ± 1.83	13.41 ± 0.56	13.13 ± 0.95	0.211	0.178	0.623
	PL/min	1.24 ± 0.16	1.22 ± 0.02	1.28 ± 0.03	1.27 ± 0.06	1.18 ± 0.01	1.23 ± 0.06	1.42 ± 0.18	1.34 ± 0.06	1.32 ± 0.09	0.208	0.176	0.618
Abalakov	Time (ms)	520 ± 7.94	417.5 ± 24.75	530.5 ± 17.68	570.67 ± 61.81	466.5 ± 51.62	493 ± 107.48	543 ± 4.36	562 ± 28.28	551 ± 12.73	0.231	0.402	0.744
	Height	33.13 ± 1	21.4 ± 2.55	34.5 ± 2.26	40.23 ± 8.91	26.85 ± 5.87	30.5 ± 13.01	36.13 ± 0.55	38.75 ± 3.89	37.2 ± 1.7	0.221	0.421	0.767
	Impulse (G)	2.02 ± 0.47	2.23 ± 0.71	2 ± 0.18	2.08 ± 0.21	2.18 ± 0.97	3.05 ± 1.62	1.7 ± 0.1	2.32 ± 0.44	2.24 ± 0.25	0.639	0.307	0.615
Multi-jump	Time Avg (ms)	473.53 ± 81.26	419.6 ± 37.9	467.5 ± 47.94	503.47 ± 32.06	484.7 ± 11.46	474.9 ± 116.39	499.27 ± 25.9	461.2 ± 60.81	511.2 ± 1.13	0.545	0.380	0.868
	Height Avg	28.03 ± 8.93	22.43 ± 4.94	27.37 ± 4.85	31.39 ± 3.69	29.49 ± 0.44	30.76 ± 10.38	30.74 ± 3.11	26.56 ± 7.04	32.67 ± 0.72	0.550	0.446	0.805
	Impulse Avg (G)	3.02 ± 0.65	4.1 ± 0.79	3.65 ± 1.11	3.8 ± 0.37	3.98 ± 1.6	3.08 ± 0.17	3.23 ± 0.4	4.25 ± 0.86	2.48 ± 0.27	0.147	0.398	0.280
RSA	Time Avg	3.84 ± 0.51	4.02 ± 0.96	4.13 ± 0.94	4.45 ± 0.39	4.59 ± 1.12	4.42 ± 0.63	3.96 ± 0.69	4.48 ± 0.34	4.13 ± 0.59	0.039 *	0.528	0.032 *
	Impacts	68.67 ± 15.5	52.5 ± 34.65	99.5 ± 21.92	61.67 ± 7.64	58 ± 39.6	88 ± 4.24	67 ± 30.05	53 ± 4.24	72.5 ± 23.33	0.434	0.474	0.236
	PL	2.76 ± 0.19	2.42 ± 0.25	2.76 ± 0.31	2.36 ± 0.16	2.25 ± 0.83	2.74 ± 0.4	2.55 ± 0.54	2.31 ± 0.09	2.45 ± 0.29	0.315	0.543	0.048 *
	PL/min	1.33 ± 0.04	1.16 ± 0.13	1.32 ± 0.16	1.14 ± 0.05	1.06 ± 0.4	1.3 ± 0.19	1.3 ± 0.27	1.18 ± 0.04	1.26 ± 0.15	0.224	0.571	0.194
*T* Test Generic	Time	14.87 ± 1.5	14.3 ± 0.13	14.66 ± 0.83	14.46 ± 0.87	13.18 ± 0.74	14.26 ± 0.42	14.51 ± 0.7	12.67 ± 0.42	12.96 ± 0.01	0.015 *	0.020 *	0.012 *
	HR Max	118 ± 22.72	168.5 ± 9.19	141 ± 19.8	168 ± 13	180 ± 9.9	162.5 ± 7.78	119.33 ± 13.32	153 ± 2.83	141 ± 14.14	0.055	0.114	0.490
	HR Avg	109.33 ± 18.58	146 ± 1.41	127.5 ± 12.02	154 ± 16.64	154.5 ± 2.12	139.5 ± 0.71	110.33 ± 21.08	143 ± 4.24	127.5 ± 17.68	0.079	0.103	0.698
	Impacts	14.67 ± 4.73	6.5 ± 2.12	29.5 ± 4.95	21.33 ± 2.52	10.5 ± 0.71	28.5 ± 2.12	20 ± 11	10 ± 7.07	19 ± 5.66	0.678	0.150	0.216
	PL	0.74 ± 0.03	0.63 ± 0.03	0.85 ± 0.01	0.78 ± 0.04	0.71 ± 0.05	0.91 ± 0.04	0.67 ± 0.08	0.51 ± 0.05	0.6 ± 0.08	0.546	0.048 *	0.028 *
	PL/min	2.9 ± 0.16	2.45 ± 0.02	3.24 ± 0.17	2.55 ± 0.22	2.34 ± 0.26	2.99 ± 0.06	2.76 ± 0.29	2.4 ± 0.15	2.78 ± 0.4	0.624	0.936	0.432
*T* Test Specific	Time	15.67 ± 1.16	15.8 ± 0.75	15.85 ± 1.93	16.85 ± 1.41	16.18 ± 1.04	16.85 ± 0.42	16 ± 1.3	14.88 ± 1.3	14.88 ± 1.3	0.390	0.058	0.145
	HR Max	131 ± 15.13	174 ± 12.73	175 ± 8.49	165 ± 24.25	179 ± 11.31	145.5 ± 33.23	144 ± 9.17	150 ± 2.83	149 ± 4.24	0.127	0.056	0.504
	HR Avg	119.33 ± 6.35	155.5 ± 0.71	161.5 ± 3.54	147.67 ± 18.77	150.5 ± 6.36	134.5 ± 19.09	131.1 ± 13.11	138 ± 1.41	135 ± 5.66	0.052	0.084	0.283
	Impacts	6 ± 1.73	6 ± 2.83	10.5 ± 2.12	10.33 ± 4.04	12.5 ± 3.54	14.5 ± 7.78	19.33 ± 21.36	5.5 ± 3.54	14 ± 8.49	0.497	0.125	0.876
	PL	0.66 ± 0.03	0.64 ± 0.02	0.77 ± 0.02	0.8 ± 0.06	0.76 ± 0.02	0.8 ± 0.04	0.6 ± 0.03	0.52 ± 0.04	0.59 ± 0.07	0.016 *	0.036 *	0.113
	PL/min	2.34 ± 0.09	2.25 ± 0.06	2.46 ± 0.18	2.42 ± 0.36	2.26 ± 0.18	2.55 ± 0.06	2.48 ± 0.21	2.4 ± 0.18	2.74 ± 0.31	0.244	0.703	0.618
Arc Test Right	Time	4.78 ± 0.06	5.09 ± 0.91	4.62 ± 0.17	5.73 ± 0.74	5.06 ± 0.76	5.69 ± 0.26	6.68 ± 0.95	6.7 ± 0.12	6.04 ± 0.69	0.070	0.072	0.170
	Impacts	16.67 ± 4.51	12.5 ± 7.78	24.5 ± 10.61	9.33 ± 0.58	13 ± 5.66	18.5 ± 3.54	13 ± 8.72	12 ± 0	22 ± 1.41	0.283	0.738	0.560
	PL	0.41 ± 0.05	0.39 ± 0.09	0.48 ± 0.04	0.39 ± 0.03	0.38 ± 0.08	0.41 ± 0.02	0.44 ± 0.1	0.4 ± 0.04	0.47 ± 0.05	0.849	0.653	0.328
Arc Test Left	Time	4.77 ± 0.27	5.04 ± 0.47	4.77 ± 0.01	6.06 ± 0.74	5.68 ± 0.72	5.44 ± 0.05	6.99 ± 0.97	5.99 ± 0.62	6.04 ± 0.56	0.026 *	0.378	0.105
	Impacts	14.33 ± 9.29	15 ± 9.9	23 ± 1.41	11.33 ± 2.08	17.5 ± 6.36	20.5 ± 6.36	17 ± 8.72	12.5 ± 3.54	18.5 ± 0.71	0.843	0.702	0.614
	PL	0.43 ± 0.03	0.39 ± 0.07	0.47 ± 0.01	0.41 ± 0.06	0.46 ± 0.01	0.41 ± 0.03	0.45 ± 0.11	0.37 ± 0.08	0.46 ± 0.02	0.913	0.075	0.275

G: guards; F: forwards; C: centers; Aerobic Test SIG/AER: aerobic capacity; Anaerobic Test SIG/ANA: anaerobic capacity; Abalakov: maximal strength test of lower body; Multi-jump: reactive strength test of lower body; RSA: repeat sprint ability test; *T* Test Generic: conventional T test; T Test Specific: specific T test with ball; Arc Test Right: right centripetal strength test; Arc Test Left: left centripetal strength test; Circuit: parts of circuit completed by the players; HR Avg: heart rate average; HR Max; heart rate maximum; HR Rec: heart rate recovery (two minutes after the end of the test); PL: player load; PL/min: player load/minute; * *p* < 0.05.

**Table 2 sports-08-00097-t002:** Descriptive and inferential results of the tests carried out according to game position.

		Guards	Forwards	Centers			
		*Avg* ± *SD*	*Avg* ± *SD*	*Avg* ± *SD*	*F*	*Sig.*	
Aerobic Capacity	Circuit	132.13 ± 7.18	132.17 ± 10.59	132 ± 10.42	0.011	0.989	
	Shots	11.13 ± 0.64	11.33 ± 0.82	11.29 ± 0.95	0.37	0.696	
	Scores	4.5 ± 2.83	2.67 ± 1.51	6 ± 2	3.956	0.038 *	∏
	Efficacy (%)	40.1 ± 24.68	23.16 ± 12.01	52.73 ± 15.46	4.255	0.031 *	Ω
	HR Avg	177.13 ± 24.45	183 ± 13.37	179.29 ± 7.67	0.188	0.831	
	HR Max	187.38 ± 25.04	192 ± 10.37	196.43 ± 8.89	0.366	0.698	
	HR Rec	133 ± 14.49	126.83 ± 14.22	137.43 ± 8.56	0.861	0.439	
	Impacts	480.13 ± 123.25	423.67 ± 170.36	641.43 ± 149.68	3.819	0.041 *	∏
	PL	26.78 ± 3.11	25.72 ± 2.16	27.62 ± 1.28	0.941	0.408	
	PL/min	2.24 ± 0.26	2.14 ± 0.18	2.3 ± 0.11	0.876	0.433	
	Acc/min	32.66 ± 4.43	33.25 ± 3.97	34.5 ± 4.06	0.048	0.953	
	Dec/min	32.65 ± 4.44	33.24 ± 3.94	34.49 ± 4.08	0.047	0.954	
Anaerobic Capacity	Circuit	113.57 ± 3.64	118.5 ± 2.95	117.83 ± 7.08	4.665	0.023 *	Ω
	Shots	29.29 ± 0.76	29.83 ± 0.75	31.17 ± 1.6	5.006	0.019 *	§
	Scores	23.86 ± 4.3	24.5 ± 2.88	28.5 ± 2.66	0.817	0.457	
	Efficacy (%)	81.41 ± 14.07	82.1 ± 9.19	91.51 ± 8.04	0.255	0.778	
	HR Avg	174.29 ± 8.04	179.5 ± 8.07	166 ± 4.86	4.022	0.036 *	§
	HR Max	192.71 ± 6.05	193.67 ± 9.67	189.33 ± 6.25	0.741	0.49	
	HR Rec	130.86 ± 17.91	137 ± 5.02	130.67 ± 10.15	0.588	0.566	
	Impacts	370.14 ± 73.54	252.5 ± 73.36	300.67 ± 45.72	0.942	0.408	
	PL	12.97 ± 1.7	12.46 ± 0.8	12.69 ± 0.61	0.613	0.553	
	PL/min	1.3 ± 0.17	1.25 ± 0.08	1.27 ± 0.06	0.618	0.55	
Strength Lower Body	Time (ms)	544.5 ± 40.82	475.83 ± 62.32	533 ± 55.21	2.42	0.117	
	Height	36.51 ± 5.82	28.15 ± 7.28	35.13 ± 6.71	2.208	0.139	
	Impulse (G)	1.95 ± 0.33	2.21 ± 0.6	2.37 ± 0.83	1.321	0.292	
Reactive Strength	Time (ms)	494.24 ± 47.15	451.93 ± 41.15	484.53 ± 60.06	1.008	0.385	
	Height Avg	30.24 ± 5.26	25.88 ± 4.79	30.27 ± 5.67	1.199	0.324	
	Impulse Avg (G)	3.27 ± 0.57	4.23 ± 0.72	3.07 ± 0.74	3.473	0.053	
Speed	Time Avg	3.92 ± 0.59	3.374 ± 1.74	3.926 ± 0.57	0.444	0.648	
	Impacts	64.86 ± 20.24	54.5 ± 23.76	88.83 ± 14.97	4.094	0.034 *	∏
	PL	2.55 ± 0.39	2.33 ± 0.4	2.7 ± 0.24	1.395	0.273	
	PL/min	1.24 ± 0.19	1.13 ± 0.2	1.32 ± 0.11	1.524	0.245	
Generic Agility	Time	14.89 ± 1.8	13.72 ± 2.53	13.82 ± 2.18	0.189	0.83	
	HR Max	137.14 ± 32.82	167.17 ± 13.6	144.33 ± 17.77	3.788	0.042 *	Ω
	HR Avg	125.29 ± 31.67	147.83 ± 5.78	128.5 ± 11.08	2.594	0.102	
	Impacts	18.29 ± 7.85	9 ± 3.85	26.5 ± 4.68	11.786	0.001 *	Ω-§
	PL	0.74 ± 0.08	0.61 ± 0.1	0.82 ± 0.1	4.252	0.031 *	Ω
	PL/min	2.65 ± 0.21	2.39 ± 0.14	3.09 ± 0.14	9.954	0.001 *	Ω
Specific Agility	Time	16.21 ± 2.17	15.29 ± 2.96	15.48 ± 2.56	0.147	0.865	
	HR Max	138.86 ± 32.78	167.67 ± 15.87	150.17 ± 27.49	3.818	0.041 *	Ω
	HR Avg	125.29 ± 26.78	148 ± 8.58	139.17 ± 21.08	4.132	0.033 *	Ω
	Impacts	13.29 ± 13.95	8 ± 4.34	12.83 ± 5.78	0.517	0.605	
	PL	0.7 ± 0.1	0.64 ± 0.11	0.73 ± 0.08	1.019	0.381	
	PL/min	2.44 ± 0.25	2.3 ± 0.14	2.55 ± 0.24	3.16	0.067	
Centripetal Force (R)	Time	5.62 ± 1.03	5.59 ± 0.96	5.64 ± 0.85	0.16	0.853	
	Impacts	11.75 ± 4.8	14.33 ± 6.06	20.29 ± 6.37	5.728	0.012 *	§
	PL	0.4 ± 0.05	0.41 ± 0.08	0.44 ± 0.05	1.789	0.196	
Centripetal Force (L)	Time	5.95 ± 1.23	5.63 ± 0.66	5.44 ± 0.57	0.685	0.517	
	Impacts	13.38 ± 6.86	16.83 ± 5.78	19.14 ± 5.18	2.368	0.122	
	PL	0.43 ± 0.07	0.43 ± 0.04	0.42 ± 0.06	0.735	0.493	

*Circuit*: parts of circuit completed by the players; *HR Avg*: heart rate average; *HR Max*; heart rate maximum; *HR Rec*: heart rate recovery (two minutes after the end of the test); PL: player load; PL/min: player load/minute; *Ω*: significant post hoc differences between guards and forwards; *§*: significant post hoc differences between guards and centers; *∏*: significant post hoc differences between forwards and centers. * *p* < 0.05.

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
