# Peer review of "Evolution of Physical Fitness in Formative Female Basketball Players: A Case Study"

_sports, 2020, doi:10.3390/sports8070097_

Round 1

Reviewer 1 Report

Thank you for the opportunity to review "Evolution of Physical Fitness in Formative Female Basketball Players: A Case Study". This type of research is definitely needed to give contributions to the basketball training process.

Overall comments
1 – Authors must review the overall English language of the paper. The sentences are very long and some expressions lose the meaning of the main idea that the authors want to transmit. I think the text need to be revised by an English native speaker in order to amend some sentences, reduce some redundancies and to clarify some expressions and ideas.

2 - As a case study, authors must make a more concise description of the case. Why does the study is focused on this specific case? What features does this case have that justify the approach done? Even stating (as is written in the point 5.2) that the results are not generalized, what is the relevant contribution of the paper to the knowledge about the game or the evaluation of the players?
The relevance of the study should be reinforced in the introduction of the paper to make clear the reasons for the interest in this specific case.

3 – All the findings of the present paper are related with the standpoint of the players and with the training process they are submitted in terms of coaching philosophy, training organization and training methodology. As the comment above the specific case must be characterized in order to have a better understanding of the different results discussed.

Focused comments

Abstract
1 – I suggest the re-written of the abstract: simplify it and try to focus it. Two examples of two confused and long sentences:
Line 12-14: “The purpose of this study was then to analyze the physical fitness of a female U18 team divided by game positions by different moments in the same season, to identify the differences and to know about the evolution of physical fitness.” The goals of the paper must be simplified.
Line 14-16: “To assess physical fitness, tests of aerobic and anaerobic capacities, lower body strength, centripetal force, agility and speed that are part of the SBAFIT battery were performed, using inertial devices equipped with microtechnologies for data collection.”

The ideas are not clear. It would be better for the article if those sentences were shorter and divided in two periods.

Methods
1 – when the participants are described the information about the basketball experience of the team elements can give additional references to make a better interpretation of the results. There is not any kind of references about the basketball experience of the sample - years of practice, hours of practice per week, eventually the specific volume of physical fitness and the specific training volume of specific basketball contents).

2 – My question is: Is it possible to make a better redefinition of the variables on the 2.3 point and try to proceed to a better systematization of the point?

Line 110; “i) Technical-Tactical Variables” are defined using the same expression.

All the other variables are “Objective”. Is it really necessary to define all the other variables with the “Objective” term? In order to simplify the terminology used in this point, authors may think about it.

Objective Internal Load Variables; Objective External Load Kinematics Variables related to Distance or Time; Objective External Load Kinematics Variables related to Accelerometry; Objective External Load Neuromuscular Variables.

It seems there are three different variables: the so-called Technical-Tactical Variables, the Internal Load Variables and External Load ones. This last group is divided in three types: Distance or Time, Accelerometry and Neuromuscular parameters.

3 – line 166/167: “(Mean and ST)” these terms are used to express the values of the descriptive parameters: mean or average and standard deviation; the more appropriate abbreviation must be m (mean) or avg (average) and sd from standard deciation. ST does not make sense.

Results
1 - the data that are supporting the radar charts are not understandable. The chart reading make us understand that there are a percentile scale describing the test performances by positions in each assessment. Where are the normalization of these data? Are they coming from the z-scores?

2 – all the tables must be revised. My suggestion is to use the terminology avg (+/-sd).

3 – The statistical results are described in the test only with the p value. The results of the statistical tests must be reported with F value, the correspondent freedom degree that are associated and finally the p value that was achieved. The right mode to report the statistic test result is: F(df)=x; p<=0.05

Discussion
1 – the discussion of the results can not be made without taking account the presage characteristics of the players and the training methodology they are submitted.

2 - To help the reader, the authors should report similarities/differences between their sample and previous studies even comparing with unspecific tests in the quantitative aspects of the measured variables. This aspect is particularly relevant because in some countries/clubs, the selection process might/might not be based on anthropometry or other performance capability of players.

Practical applications
1 - Based on their findings, specific and gender-related practical applications should be provided to the coaches. In particular, which specific tasks/training sessions could be suggested to improve speed, agility and explosive strength in female in relation to the actual training plan (e.g., number and duration of sessions)

Reviewer 2 Report

The main problem of the manuscript is a very small sample of participants (3G, 4F, 3C). From this point of view is difficult to compare results with ANOVA and results of statystical analysis is questionable. The suggestion to the authors is to rewrite manuscript and compare obtained results during the season for whole sample without comparing them according to the players positions. However, the whole manuscript should be in this case rewritten and whole sample should be treated like one team and results should be compared through the season for statystical analysis. In this case also training should be included to discuss why change through the season occured or not.

Reviewer 3 Report

The article deals with a case study on Evolution of Physical Fitness in Formative Female Basketball Players. It is a qualitative study carried out on a single group of athletes. It did not include a control group and this could be a limit. Considering that the research represents a "case study", the results obtained and described with descriptive and inferential statistics, can be considered sufficient.

Reviewer 4 Report

The study analyzed physical fitness of female basketball players by game position and different time points in a season. The topic is interesting and important. The manuscript is well written overall.

Meanwhile, the statistical analysis in the study needs some clarifications. The whole discussion and conclusions in the manuscript depend on whether the correct statistical analysis was performed. I would like to see that the authors elaborate some aspects of the analysis, before judging the results, discussion, and conclusions. Specific comments of mine are listed below.

Line 97: “… (n=30).”
Please be sure that the sample size is still n = 10, whereas it can be said that the total number of data points is 30.

Line 168: “…, finding a normal distribution of data, so parametric test were carried out …”
I am not so sure if this normality analysis was meaningful based on the small sample size (n = 10; 30 data points in total). This is especially the case for the analysis based on game position (n = 3-4 for each position). Because of the small sample size, I suggest that the authors do not rely on this analysis and hence normality assumption; rather, the authors use alternative methods, such as permutation tests, to produce more robust findings.

Line 170: “… one-way ANOVA with Bonferroni Post Hoc both by assessment and global with all the data obtained …
Please clarify what all the data above mean. If all the data included repeated-measures data, then one-way ANOVA would not likely be appropriate, since independence of observations, probably the most important assumption, was not met. Repeated-measures data collected from the same subjects are likely correlated, and appropriate statistical analysis techniques, such as repeated-measures ANOVA, should be used to analyze such data. Please clarify this aspect of the analysis, and perform the correct analysis, if needed. If my understanding here is not correct, please clarify the analysis as well.

Line 175: “The Mauchly’s W sphericity test was …”
It is known that Mauchly’s W is under-powered with a small sample size, and over-detects the departure from normality with a large sample size. My suggestion is to not rely on Mauchly’s W; rather, the authors conduct the analysis with corrected statistics, assuming that the assumption of sphericity was not met. If the authors have different ideas, please specify such ideas in the manuscript.

For the overall analysis:
This was mentioned above. Because of the small sample size, I suggest that the authors use alternative, more robust statistical analysis techniques, such as permutation tests which work well for small data and produces exact/Monte Carlo p-values.

Figure 1:
I believe that the values in the figures are standardized ones. In the main text, Line 178, it is said that z-scores were used to standardize the values. Z-scores have a mean of zero and standard deviation of 1. The values in Figure 1 do not seem to be z-scores. Please clarify it.

Table 1: Column, W de Mauchly:
I am not sure if the column of W de Mauchly in the table adds anything to the results. It is the test for sphericity assumption, and I don’t think that this column is really necessary. Please clarify it.

Results section overall:
This depends on the style of a journal, but the results section in this manuscript is written in the present tense. It is normally written in the past tense. Please re-write the section using the past tense, unless the journal specifically asks that this section be written in the present tense.

Round 2

Reviewer 2 Report

After mayor revision and clarifications of the authors now the paper is ready to be accepted in present form.

Reviewer 4 Report

The authors have addressed in the revised manuscript the aspects of the report that were unclear to me in the original version. I believe that the manuscript is now qualified for publication in Sports.